# Protocol for establishing a child and adolescent twin register for mental health research and capacity building in Sri Lanka and other low and middle-income countries in South Asia

Kaushalya Jayaweera [1] Jeffrey M Craig,[2] Helena M S Zavos,[3] Nihal Abeysinghe,[1] Sunil De Alwis,[4] Alina Andras,[5] Lasith Dissanayake [1] Krysia Dziedzic,[6] Buddhika Fernando [1,5] Nick Glozier,[7] Asiri Hewamalage,[4] Jonathan Ives,[8] Kelvin P Jordan,[9] Godwin Kodituwakku,[1] Christian Mallen,[10] Omar Rahman [11] Shamsa Zafar,[12] Alka Saxena,[13] Fruhling Rijsdijk,[14] Richard Saffery,[15,16] Emily Simonoff,[17] Rita Yusuf [18] Athula Sumathipala,[1,5] We would like to acknowledge the other members of the SEARCH Group

For numbered affiliations see end of article.

**Correspondence to**
Professor Athula Sumathipala;
a.sumathipala@keele.ac.uk

## ABSTRACT

**Introduction** Worldwide, 10%–20% of children and adolescents experience mental health conditions. However, most such disorders remain undiagnosed until adolescence or adulthood. Little is known about the factors that influence mental health in children and adolescents, especially in low and middle-income countries (LMIC), where environmental threats, such as poverty and war, may affect optimal neurodevelopment. Cohort studies provide important information on risks and resilience across the life course by enabling tracking of the effects of early life environment on health during childhood and beyond. Large birth cohort studies, including twin cohorts that can be aetiologically informative, have been conducted within high-income countries but are not generalisable to LMIC. There are limited longitudinal birth cohort studies in LMIC.

**Methods** We sought to enhance the volume of impactful research in Sri Lanka by establishing a Centre of Excellence for cohort studies. The aim is to establish a register of infant, child and adolescent twins, including mothers pregnant with twins, starting in the districts of Colombo (Western Province) and Vavuniya (Northern Province). We will gain consent from twins or parents for future research projects. This register will provide the platform to investigate the aetiology of mental illness and the impact of challenges to early brain development on future mental health. Using this register, we will be able to conduct research that will (1) expand existing research capacity on child and adolescent mental health and twin methods; (2) further consolidate existing partnerships and (3) establish new collaborations. The initiative is underpinned by three pillars: high-quality research, ethics, and patient and public involvement and engagement (PPIE).

**Ethics and dissemination** Ethical approval for this study was obtained from the Ethics Review Committee of Sri Lanka Medical Association and Keele University's Ethical

## Strengths and limitations of this study

► The only population-based register of child and adolescent twins established in a low and middle-income country.
► The project includes database development, ethics development and public and patients involvement and engagement.
► The cohort compliments the existing adult twin register in Colombo Sri Lanka and 15 years of twin research experience.
► The focus on Sri Lankan twins may limit the generalisability of some findings to other populations.

Review Panel. In addition to journal publications, a range of PPIE activities have been conducted.

## BACKGROUND

A substantial proportion of the world's health problems in both high-income countries (HICs) and low and middle-income countries (LMICs) arises from mental, neurological and substance use conditions.[1] Worldwide, 10%–20% of children and adolescents experience mental health conditions, and many transition into adulthood undiagnosed and continue to experience these disorders.[2] Child and adolescent mental health is especially important in LMIC, as treatment and prevention resources are generally low in comparison to adults.[3] A number of prenatal, perinatal and postnatal factors have been associated with increased risk of neurodevelopmental disorders and mental health

conditions across the life course.[4–6] However, much of this research has been based on HIC. By establishing a child and adolescent twin cohort, we will establish a resource that will enable studies to conduct research on the impact of early life risk factors on the development of neurodevelopment and mental health conditions in an LMIC.

Challenges to early brain development take place in intrauterine life and continue throughout childhood and adolescence. Research has shown a number of key risk factors in the fetal and early postnatal life as being associated with a range of neurodevelopment and mental health conditions.[4] Research in HICs has suggested that nutrients and growth factors regulate brain development during fetal and early postnatal life, and rapidly developing brain is more vulnerable to nutrient insufficiency.[7] Other factors such as low birth weight and hypoxia, which are more common in LMICs,[8] are among the environmental factors that have been consistently associated with a range of neurodevelopmental and mental health conditions across the lifespan.

Disasters, both natural and war, are well-known causes of a variety of psychological and psychiatric conditions.[9] Research has suggested that children and adolescents exposed to war-related traumatic events are affected mental health distress and in certain cases long-term psychopathology.[10] There is also a strong relationship between adolescents exposed to trauma and suffering from Post Traumatic Stress Disorder (PTSD) and substance abuse. Studies have shown that more than half of the young people with PTSD subsequently develop substance abuse problems.[11] These publications have identified the need for more research in LMICs.

WHO, in a survey of mental health research priorities in LMICs including Asia, identified children and adolescents exposed to violence/trauma as prioritised population groups,[12] while risk and resilience can impact on Mental, Neurological and Substance Use Disorders (MNS) disorders and their outcomes. In both cases, there is a significant gap in research in LMICs.[13] Interestingly, there are relatively low rates of common mental disorders among adults in Sri Lanka.[14] However, limited similar research has been conducted to date on children and adolescents in Sri Lanka.

Cohort studies, in which groups of people are followed up over time, are the best method to determine disease incidence and longitudinal pattern of physical and mental health conditions and the natural history. Cohort studies can provide important information on risks and resilience factors across the life course. They can also provide data on childhood, parental and early life environmental factors associated with poor health during childhood and in later life, and risks and resilience factors across the life course. Large longitudinal birth cohort studies have been conducted within HIC, but are not generalisable to LMIC.[15] In order to reduce MNS disorders in adolescents in LMIC, Davidson and colleagues[13] recommend increased support and development of longitudinal cohort studies. Only a limited number of LMIC birth cohort studies exit and are located in Guatemala, India, Brazil and Philippines.[13] Findings from these studies are difficult to apply to a country such as Sri Lanka due to significant sociocultural differences; further cohorts throughout South Asia are required to generate data to form new strategies to reduce the incidence and impact of MNS disorders in the region.

The use of the co-twin control design (using differences within twin pairs to examine the association between a putative environmental risk factor and an outcome variable) is extremely efficient for examining risk factor–outcome associations compared with unmatched designs. And through intervention and non-intervention studies offers many novel utilities for cutting edge research: genomics, genetic approaches for prevention,[16] therapy genetics[17] and studying genetic determinants of response to pharmacological, as well as psychotherapeutic treatment.[18] Involving twins in randomised control trials (RCT) and randomising concordant monozygotic (MZ) twins to two arms can control for many of the potential confounding factors, especially genetic makeup due to matching.[19] It permits large-scale epidemiological studies, combining twins with age/gender-matched non-twin samples, of both mental disorders and normotypic behaviour.

In longitudinal observational studies, confounding poses a considerable threat to the validity of studies aiming to identify causal mechanisms. Twin cohorts can be useful as they allow researchers to understand the role of genes and environment on outcomes of interest. In addition, a number of studies have found evidence of genetic influences on environmental variables such as stressful life events,[20] a phenomena know as gene-environment correlation.[21] This additional aetiological information can help us to understand more about the causal relationships, phenotypes and environments, and phenotypes of interest. As twinning is largely unrelated to environmental risk factors such as poverty or social class, twin studies allow generalisable assessments of associations and the ability to evaluate the extent of both genetic and environmental confounding.[22]

### Focus on Sri Lanka

Anthropologically, Sri Lanka has a genetically diverse population of 20 million with a high literacy rate and with 33.3% of the population below 18 years of age. It has endured a three-decade civil war and the Asian Tsunami in 2004. Ten years post-war, it is experiencing demographic transition, rapid globalisation, urbanisation and internal/external migration.[23]

The majority of the Sri Lankan population lives in the rural sector (77.4%); the urban population share is 18.2%, while 4.4% live in large agricultural estates, considered a separate sector.[24 25] Poverty is a major problem and education levels are relatively lower in rural areas.[25] Vavuniya (Northern Province) has been chosen as a comparative setting to Colombo (Western Province) as it is a more rural region directly affected by civil war. Urban/rural

differences can have an impact on the onset of mental health,[25] and therefore, we included these two areas to develop the register initially.

## Aims

The overall objective is to extend the health and social care research carried out in Sri Lanka, especially by the Institute for Research and Development in Health and Social Care, Colombo (IRD). Our current twin research follows a 9500 twin pair cohort of adults[14] in the population-based Sri Lankan Twin Registry.[26] The IRD has a strong core team with a variety of expertise and has assembled a team of long-term collaborators; local, regional and from HICs who have expertise in the three main pillars of this initiative which are high-quality research, ethics and PPIE. We aim to (1) establish a register of infant, child and adolescent twins, and mothers pregnant with twins in the districts of Colombo and Vavuniya, to be approached for future research projects; (2) gather preliminary information needed to develop future grant applications; (3) initiate and conduct patient and public involvement and engagement (PPIE) work in Sri Lanka through the organisation of local events such as workshops and meetings with stakeholders and (4) bring together investigators from key long-term partner organisations in LMICs and HICs and establish the planning and coordinating units, and steering committee. Research, ethics and patient and public involvement and engagement (PPIE) expert groups will be developed to strategise and plan relevant activities.

## METHODS

The broad hypothesis of this project is to apply classical twin methods and the MZ differences design to understand the prevalence and aetiology of a range of neurodevelopmental and mental health conditions affecting children and adolescents in two districts of Sri Lanka. Specific research projects to be undertaken will be decided by the IRD in consultation with relevant stakeholders in Sri Lanka and supported by the methodology experts group of SEARCH based on available funding calls.

### Study setting: Colombo and Vavuniya districts of Sri Lanka

Colombo district is predominantly urban and comprises 10% of the Sri Lankan population. The ethnic distribution is 76% Sinhala, 12% Tamil and 11% Moor; 15% of the district population is concentrated in the greater Colombo area where the ethnic distribution is almost equal in proportion to each other. The Vavuniya district is mainly classified as rural, and the ethnic majority are Tamil (83%).[27] These new cohorts provide a good ethnic and urban/rural mix. The proposed new register will complement and supplement the exiting adult twin register in Colombo district.

### Inclusion and exclusion

All twins below the age of 18 years and mothers pregnant with twins living in Colombo and Vavuniya districts will be eligible and will have the same opportunity to be registered or not to be registered, based on their or their parents' or guardians' (for minors) wishes.

### Approaches to identification and recruitment

There will be no recruitment for specific research but only registration onto the database. Twins, and their parents, and mothers pregnant with twins will be contacted for consent to be included in the register. Database registration will not automatically mean involvement or mandatory participation in research.

All future projects will be first reviewed by the EEG in SEARCH project before proceeding to external ethical review in Sri Lanka and the country of external collaboration. To recruit twins and to maximise the outcome of this feasibility study, we decided on four source populations detailed below:

### Existing twins from adult twin registry

Twin offspring are more common among dizygotic twins compared with singletons and monozygotic pairs.[28] The adult population-based twin register in Colombo comprises 9500 twin pairs of which 4112 twin pairs are of childbearing age. Of the 4112, approximately 2% would be predicted to have dizygotic (DZ) twins (n=82). There are also around 1000 pairs from Colombo aged up to 18 years voluntarily registered since establishing the adult register. We will contact these adult twins by phone and postal methods to determine twin offspring aged up to 18 years and invite them to register. Ethics approval and consent has been obtained previously for contacting these twins.[29]

### Maternity hospitals

Almost all deliveries (singletons and higher order multiples) take place in major hospitals in Sri Lanka. We will determine the feasibility of using existing hospital records (in Colombo and Vavuniya) from teaching and maternity hospitals to identify newborn twins and mothers attending antenatal clinics pregnant with twins. Additionally, we will hold discussions with hospital management on feasible variables for data collection and using anonymised data from the routine health records. Potential future data to be collected are detailed in table 1.

### Field visits

We will work in collaboration with the Family Health Bureau, which is the central organisation of the Sri Lankan Ministry of Health responsible for Maternal and Child Health Services. The front-line health worker providing domiciliary care to mothers and children within the community is the Public Health Midwife (PHM). They work within a well-demarcated area having a population ranging from about 2000 to 5000 and conduct regular field visits. We will explore the feasibility of identifying twins by PHM to invite for registration.

**Table 1** Potential data to be collected in future projects

| | |
|---|---|
| Data from hospitals | **Data up to the date of delivery**<br>▸ Expected date of delivery and actual date delivered<br>▸ Medical and obstetrics related data (maternal complications)<br>▸ Ultrasound scan data<br>▸ Method of delivery (normal/instrumental/section)<br>**Postdelivery**<br>*Infant related*<br>▸ Placentation/zygosity<br>▸ Neonatal complications<br>▸ Birth weight/length/head circumference/Apgar score<br>▸ Weight gain<br>▸ Mile stones<br>▸ Breast feeding<br>▸ Bonding<br>*Maternal*<br>▸ Sociodemographic data<br>▸ Type of conception (normal/IVF)<br>▸ Mood (happiness/sadness)<br>▸ Maternal nutrition-related data<br>▸ Maternal mental health data<br>▸ Postnatal complications<br>*Maternal and paternal*<br>▸ Willingness to give consent for biospecimens of the infant |
| Data from primary healthcare workers in the field.<br>Data will be collected from clinics/house visits. | **Twin related**<br>*From healthcare records available*<br>▸ Birth weight/length/head circumference/Apgar score<br>▸ Weight trends over the years<br>▸ Immunisation history<br>▸ Healthcare/illness history including developmental delays<br> **Measurements done at home**<br>▸ Data through questionnaires for:<br> – Mental health of child<br> – Cognition of child<br> – Nutrition related<br> **Data from parents**<br>▸ Maternal and paternal data as mentioned above<br>▸ Qualitative interviews from twin's parents:<br> – Including beliefs, attitudes and issues about twins<br> – Challenges in rearing twins<br> – Financial implications compared with singletons |
| Data from schools | **From available school records**<br>▸ Entry behaviour assessment at Grade 1<br>▸ Academic attainments indicated through school reports<br>▸ Records of competency development of children during the primary cycle.<br>**Teacher related**<br>▸ If the twins are in the same school or class<br>▸ Engagement with peers<br>▸ Mental health of child |

IVF, in vitro fertilisation.

## Schools

With the support of the National Institute for Education and Ministry of Education Sri Lanka, we will conduct a feasibility study of tracing twins through a school survey. There are 584 government schools in both districts and will be possible to identify twins and obtain information about the number of twin pairs in each school and feasibility of recruitment and engagement. Through the schools, we will contact the parents of those twins and provide the information leaflet to register their twin children.

## Data to be collected

The primary data collected during the establishment of the register are confined to basic demographic information such as names, dates of birth and contact details.

Table 1 describes the sources and potential types of data to be collected in future studies. We will look at the potential data available from school records such as academic performance, entry behaviour assessments administered by Grade 1 on entry to school, as well as follow-up on their competencies using criteria developed by National Institute of Education, up to grade 5, and academic attainment through school reports. Additionally, we will also explore the feasibility of obtaining data such as weight trends over the years, immunisation status and any relevant data collected by the PHM. With the permission from the hospitals and pregnant mothers, we shall also assess the possibility of obtaining recorded data.

### Sample size
We will recruit as many twins as possible in both districts. The Colombo district has 40 000 births per year, with an estimated number of 7000 twin pairs under the age of 18 years, and the Vavuniya district 4000 births per year and 700 twin pairs, up to the age of 18 years.

### Testing the feasibility of establishing regional twin registers
We will also initiate feasibility/exploratory studies with our partner institutions Bangladesh, India and Pakistan. Feasibility proposals submitted by these institutions will be reviewed by the Steering Committee and experts in ethics within this group, and successful applications will be provided limited seed funding through the IRD, Colombo, through allocated funds within the MRC grant for this study. Institute for Mind and Brain was successful in obtaining seed funding to carry out an initial small-scale prevalence study in the local area of Wadakkanchery, Kerala, India, in order to estimate the incidence of infant, child and adolescent twins in the local school population. This was the first step in planning to establish a confidential twin registry in the State of Kerala. The team will also conduct a qualitative study to test the feasibility of creating the Kerala twin registry.

### Capacity building
Capacity building is a moral and ethical obligation of externally funded international collaborative research. It will also be bidirectional and collaborative, to advance the local and global agenda is a major component of the project. Ongoing and newly proposed collaborative courses and workshops conducted IRD, and this project's collaborators will be open to junior colleagues proposed by the co-investigators. We will initiate ethics and twin research-specific capacity building process in both HIC and LMIC as well, through introductory and advanced courses on twin methods, further strengthening the early and mid-career capacity enhancement by offering the opportunity to work with expert groups. Funding for such fellowships will be obtained through future grant applications. Capacity building of junior academics across the partnership will develop future research leaders and

core skills on child/adolescent mental health and twin research methods.

## ETHICS AND DISSEMINATION
An internal project governance system and an Ethics Experts Group (EEG) consisting of experts in ethics within this project team and external procedural ethics review will ensure ethical conduct of the highest standards. The key ethical sensitivities arising from future research will focus on participants being a vulnerable group (minors, some having mental illness), involving proxy consent, potential for future complex genetic and omics research, sensitive personal data storage, confidentiality, anonymity and use of and access to data, and international collaborations in a different culture.

### Managing the involvement of this vulnerable group
Carrying out research among vulnerable populations needs to be handled with extra caution due to the inherent dilemma in such research. We have taken a proactive approach by setting up the EEG to provide ethical oversight in all activities, to ensure the right blend of scientific rigour and ethics, while also providing a platform to peer review international collaborative research projects. Protection of twins'/parental interests will be ensured, for example, by avoiding undue incentives to participate, mechanisms to voice concerns, ensuring awareness of the right to withdraw. The EEG will also set up an internal system of governance to regulate the use of the registry that will work in conform with local ethical review practices.

For all future research project, specific ethics review and approval will be obtained from relevant ethics review committees (ERCs). We will obtain specific ERC approval and consent from participants for any research involving collection and storage of human material. Data collected will be pseudo-anonymised but remain linked to participants using a unique identity number.

The IRD has existing guidelines, which have been published,[30] and contain the existing policies and guidelines on access to and sharing data/material. It addresses specific concerns when involving human participants and potential use of human tissue and data collection, documentation, storage and access. These guidelines will be revised and updated accordingly. In addition, in developing the governance structure we will also incorporate the Council for International Organizations of Medical Sciences guidelines for future research: more specifically guideline 12; a governance structure in place to ensure the appropriate use of stored data that is particularly relevant to the proposed work will be considered.

### Consent and information provision
Consenting to participate in a cohort study is a long-term commitment, often an uncertain one when specific use of data is not known in advance. Including children and minors in such a project requires parental consent but

raises challenges as children grow into mature minors and adults. The EEG will give careful thought to how ongoing consent, assent and re-consenting will be undertaken as children develop into autonomous persons who can engage in, and make decisions about, research participation. Potential participants will be contacted for consent for register inclusion, but database registration does not mandate research participation; appropriate consenting processes will be developed to ensure optimal balance between respect for participants' autonomy and overburdening with individual participation requests. Participants will be informed that registration is entirely voluntary and that there is no obligation for them to take part. Incentives or payments will not be provided for registering in the twin database.

We will consult the EEG on how best to contact and obtain assent/consent from twins and their parents. Information leaflets (which have received ethical approval) will be provided to all potential participants. The leaflets will emphasise that the database registration does not involve mandatory participation in research. And that future research will need separate ethics clearance from the country of origin and Sri Lanka.

### Data storage, sharing, access confidentiality and anonymity

Existing IRD institutional policies will be followed to ensure secure data storage and usage to guarantee the obligations of confidentiality and anonymity towards participants. IRD will have ownership of the data collected in future research projects. We will only share anonymised data with academics; however, they will not have direct access to the IRD twin databases. Personal data of participants will be stored in a confidential, password-protected database accessible by only the IRD team. Any documents that contain non-anonymized data will be stored in line with UK Data Protection Act (1998). The data sharing policy based on existing institutional policies will be agreed at the outset.

Consent forms will be kept in a separate location to the research data used for analysis so that no linkage can occur. All data sets and databases will be stored on a secure, password-protected computer in IRD's secured data room. Backups of this data will be done regularly and stored along with any hard copy information in a locked filling cabinet of the data room.

### Public involvement

The pioneering work in PPIE, which is a relatively novel concept in Sri Lanka and South Asia, will provide a unique opportunity to benefit LMIC and HIC as well through reciprocal learning. Evidence from the UK demonstrates that the quality of research is better, and the likelihood of successful recruitment and implementation of the findings is improved when patients and the public are involved in research.[31]

The IRD in collaboration with the Sri Lankan Medical Association and Keele University, UK, held a 1-day pre-congress workshop at the 131st Anniversary International Medical Congress of the Sri Lanka Medical Association. Resource persons from IRD, Keele University, and other guest lectures from Sri Lanka who are experts in PPIE shared their experience and knowledge with 72 workshop participants. This workshop was preplanned and funded through MRC, UK grant funding obtained for this study.

We aim to establish a culture of PPIE in Sri Lanka, working with both academic colleagues and lay members, in this instance specifically with twins and their parents. We will recruit lay members to a PPIE group to oversee the work described and work with academics to plan and develop future research questions. Training and culturally adapted materials will enhance their engagement. Engaging more professionals/celebrities (parents of twins or twins themselves) in PPIE activities will increase awareness of twin-specific issues from a sociological perspective.

### Dissemination plan

Engagement, communication and dissemination of research and related activities beyond academia are a priority at the IRD. Since its inception, the IRD worked closely with academic partners, policymakers, the public and the electronic/print media. We will promote twin research through electronic media and print, especially using our own trilingual research and science magazine 'Gaveshana'. Gaveshana is published quarterly by the IRD and approved by the Ministry of Education for distribution to schools. Scientific publications in high impact international journals will allow the sharing of knowledge through a wider network. The initiative and the strategy to work beyond twin methods will be widely publicised to attract a wider research community.

### TARGETED OUTCOMES OF THE PROJECT AT THE END OF 2 YEARS
#### Outputs

1. A sampling frame (the register) of newborn/child/adolescent twins and mothers pregnant with twins with consent to invite for future research.
2. Identification of a core set of potential data variables, which are of sufficient quality for research and are feasible to extract from routine records on children's health and education.
3. Development of key research proposals, taking into consideration research gaps in child and adolescent mental health research.
4. Revision of the existing IRD ethics and governance guidelines.
5. PPIE-related work that will include a twin gathering event organised to initiate the process of developing networks and engaging twins in twin research.

### DISCUSSION

This unique register will increase the volume of high-quality research and generate a wealth of longitudinal data on mental health and other disciplines in a South

Asian LMIC. Phenotypic, biological and anthropometric data, physical measures and social data will allow the studying of health and well-being throughout the life course. It will also take cross-cultural research into a global context and will provide a better understanding of the mechanisms and dimensions that underlie vulnerability and resilience to mental illness, as well as the personal, social and economic impact of mental illness as a consequence of wider ranging external drives and risk factors. And popularise twin research methods provide capacity enhancement for junior academics and advances on ethics and PPIE in Sri Lanka and within the region.

This initiative provides capacity building in ethics for LMIC and UK researchers, especially on issues related to cutting edge collaborative research in LMIC. Capacity building benefit specifically on ethics will include enhanced training and knowledge on vulnerable groups (minors, those with mental illness), proxy consent and international collaborations. Academics in other disciplines will benefit in this endeavour as well. This will be two-fold: first, research beyond mental health and second beyond twin research: the collection of a wealth of longitudinal phenotypic, biological anthropometric and physical measures as well as economic, social and psychological data on families with twins. It also enables pioneering education research. The initiative has implications for fetal medicine because twins are associated with prematurity, low birth weight, greater intrauterine growth retardation, more birth defects, birth complications and neonatal mortality.

Social value arising from the project is through the potential impact on many domains. Potential societal impact will come from highlighting the additional economic, social and psychological impact of being a twin, thereby highlighting twins as a unique societal stratum and creating strong awareness among public and policy planners on the above issues.

Since there is a constellation of exposures that put adolescents at risk of neurodevelopmental and mental health conditions and many of which may be simultaneous outcomes of other exposures, research should measure multiple factors.[13] Therefore, the proposed work will offer a significant opportunity for future LMIC research in diverse disciplines.

A focus on Sri Lankan twins may limit the generalisability of some findings to other populations. The reason is because of the varying impact of environment between HIC and LMIC. Heritability is population-specific and due to genetic differences between populations, analyses need to be performed in all ethnic groups to understand a trait at a world-wide level. Being the first child and adolescent twin registry in Sri Lanka will allow to answer questions that could not be studied before. Inhabitants of LMIC are often under-represented in genetic studies, and therefore, the studies arising from this database will add significant value to the scares research in LMIC.[32]

**Author affiliations**
[1]Institute for Research and Development in Health and Social Care, Colombo, Sri Lanka
[2]Centre for Molecular and Medical Research, School of Medicine, Faculty of Health, Deakin University, Geelong, Victoria, Australia
[3]Department of Psychology, Institute of Psychiatry, Psychology & Neuroscience, Kings College London, London, United Kingdom
[4]Ministry of Health, Nutrition and Indigenous Medicine, Colombo, Sri Lanka
[5]Research Institute of Primary Care and Health Sciences, Keele University, Newcastle-under-Lyme, Staffordshire, United Kingdom
[6]Arthritis Research Campaign National Primary Care Centre, Stoke on Trent, UK
[7]Brain and Mind Centre, University of Sydney, Sydney, New South Wales, Australia
[8]Department of Population Health Sciences, Centre for Ethics in Medicine, University of Bristol, Bristol, UK
[9]Primary Care and Health Sciences, Keele University, Keele, UK
[10]Arthritis Research UK Primary Care Centre, Keele University, Keele, UK
[11]Public Health, Independent University, Dhaka, Bangladesh
[12]Centre of Excellence in MNCH, Health Services Academy, Islamabad, Pakistan
[13]Genomic Research Platform and Single Cell Laboratory, Biomedical Research Centre, Guy's and Saint Thomas' Hospitals NHS Trust, London, UK
[14]Social, Genetic, and Developmental Psychiatry Centre, Institute of Psychiatry, Psychology & Neuroscience, King's College London, London, UK
[15]Cancer and Disease Epigenetics, Murdoch Childrens Research Institute, Parkville, Victoria, Australia
[16]Paediatrics, University of Melbourne, Parkville, Victoria, Australia
[17]Child and Adolescent Psychiatry, Institute of Psychiatry, Psychology & Neuroscience, King's College London, London, UK
[18]School of Life Sciences, Independent University, Dhaka, Bangladesh

**Acknowledgements** We would like to acknowledge Dr Jayanthi Gunesekara (National Institute of Education), Mrs Jayani Tillakaratne (Ministry of Education) and Mr Saman Indrajith (The Island Newspaper, Sri Lanka). We also highly appreciate the ongoing support from Dr Nethanjalee Mapitigama, Dr Kaushalya Kasthuriarachchi and Dr Hiranya Jayawickrama of the Family Health Bureau, and Dr Ruwan Wijayamuni, Chief Medical Officer, Colombo Municipal Council. We would also like to thank higher officials of the Ministry of Health and the Ministry of Education. We gratefully acknowledge members of the Institute for Research and Development, Dr Irena Zwierska and Ms Rebecca Parker at Keele University who have contributed immensely to the preparation and execution of the project.

**Collaborators** We would like to acknowledge the other members of the SEARCH Group: Professor Aasim Ahmad (The Aga Khan University, Pakistan), Dr Achala Jayathilake (Post Graduate Institute of Medicine, Sri Lanka), Professor Ajith Nagahawatte (University of Ruhuna, Sri Lanka), Dr Anant Bhan (Bioethics and Global Health, India), Dr Chandima Wickramatilake (University of Ruhuna, Sri Lanka), Dr Gominda Ponnamperuma (University of Colombo, Sri Lanka), Dr Lavan Selvarathnam (Ministry of Health, Sri Lanka), Professor Preethi Udagama (University of Colombo, Sri Lanka), Dr Sorcha Uí Chonnachtaigh, Dr Steven Blackburn, Dr Tom Shepherd and Dr Toby Helliwell (from Keele University, UK). Dr Sudath Samaraweera (Ministry of Health, Sri Lanka), Dr Thamasi Makuloluwa (General Sir John Kotelawala Defence University, Sri Lanka), Dr Yatan Balhara (All India Institute of Medical Sciences, Delhi) and Ms Ruwini Cooray (Institute for Research and Development).

**Contributors** All authors contributed to the drafting of original funding application, which is the basis of this protocol paper. AS was responsible for the initial draft of the overall concept for the application and the paper; FR, RS, ES, JMC and HMSZ were mainly responsible for twin design; BF, JI and AS were mainly responsible for ethics and governance component; NA, NG, APH, KPJ, CM, OR, SZ, RY, KJ and LD were mainly responsible for epidemiology and statistics components; SBDA, KD and GK were mainly responsible for PPIE component; AS, for genetic component; LD and AA contributed to obtaining ethical clearance. KPJ, LD and AS contributed to all stages of planning and writing/editing the article based on the original funding application. As this is a protocol paper, there was no acquisition of data or analysis and interpretation of data. All authors contributed to different drafts of the paper and finally approved the manuscript for submission.

**Funding** This study is supported by a 1-year pump priming grant awarded by the Medical Research Council, UK (Grant number: MC_PC_MR/R018448/1). AS currently receives salary support in kind from the Midland Partnership Foundation NHS Trust, UK.

**Competing interests** None declared.

**Patient consent for publication** Not required.

**Ethics approval** Ethical approval for development of a child and adolescent twin registry was obtained from the Ethics Review Committee of Sri Lanka Medical Association (ref no: ERC/17-025) and Keele University's Ethical Review Panel (ref no: ERP3157).

**Provenance and peer review** Not commissioned; externally peer reviewed.

**ORCID iDs**

Kaushalya Jayaweera http://orcid.org/0000-0003-3780-0901
Lasith Dissanayake http://orcid.org/0000-0002-3609-4358
Buddhika Fernando http://orcid.org/0000-0001-6352-2832
Omar Rahman http://orcid.org/0000-0002-0606-2507
Rita Yusuf http://orcid.org/0000-0002-0151-9571

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
