## [Reviewer comments · BMJ Open]

ARTICLE DETAILS

TITLE (PROVISIONAL)	Protocol for establishing a child and adolescent twin register for mental health research, and capacity building in Sri Lanka and other low and middle-income countries in South Asia
AUTHORS	Jayaweera, Kaushalya; Craig, Jeffrey; Zavos, Helena; Abeysinghe, Nihal; De Alwis, Sunil; Andras, Alina; Dissanayake, Lasith; Dziedzic, Krysia; Fernando, Buddhika; Glozier, Nick; Hewamalage, Asiri; Ives, Jonathan; Jordan, Kelvin; Kodituwakku, Godwin; Mallen, Christian; Rahman, Omar; Zafar, Shamsa; Saxena, Alka; Rijdsdijk, Fruhling; Saffery, Richard; Simonoff, Emily; Yusuf, Rita; Sumathipala, Athula

VERSION 1 – REVIEW

REVIEWER	Gennaro Catone Suor Orsola Benincasa University
REVIEW RETURNED	20-Apr-2019

GENERAL COMMENTS	I read with interest the paper entitled "Protocol for establishing a child and adolescent twin register for research and capacity building on the aetiology of mental illness in Sri Lanka and other low and middle-income countries in South Asia". The topic is very important. Longitudinal studies are necessary to better understand mental health problems among children and young adults. The merit of this study is to perform a twin register in Sri Lanka (low-middle income country) and authors rightly reiterate (in the abstract and in the text) that studies in these contexts are fundamental for environmental factors that are different from more industrialized countries. Below there are some suggestions to improve the manuscript: 1) The English language must be completely revised by a native speaker 2) BACKGROUND in my opinion this section should be revised. More information should be provided: a) the multifactorial aetiology of mental health disorder that includes also environmental factors; b) the need of preventive actions on these environmental factors. Information are presented too briefly and disconnected from each other. -page 6 line 85-86. Authors stated that reasons may include... but they cited only one reason "the stigma", in my opinion they have to discuss other factors. -page 6 line 88, insert also impact of mental illness on well being.
---

	-lines 83-93. I suggest to revise this section. I will first proceed by stating that the etiopathogenesis of mental disorders in the developmental age is multifactorial, then I will go on to say that in the LI countries it is worth investigating environmental factors (different from HI countries) and then I would mention the information of the WHO. -lines 94-100. Authors affirmed that studies in adolescence are scarce. In my opinion authors should reinforce this affirmation saying why adolescence is important (the onset of several mental disorder is in adolescence; adolescence is a critical developmental psychological period). -lines 101-108. Authors should introduce the concept of resilience and then use the term -The impact of the challenges to early brain development on mental health and on the etiology of mental illness. I do not understand this paragraph. If author would expose neurodevelopment they should refer not only to pregnancy period and perinatal period and not only to schizophrenia. Rationale is that neurodevelopment is not only intrauterine or perinatal period and not only linked to schizophrenia. The chapter is really incomplete and not linked with background and subsequent paragraphs. I believe that if the authors want to keep this paragraph they must proceed with an extensive narrative revision of the topic otherwise it is better to delete the section - Why cohort studies?. Could author briefly explain the characteristics of cohort studies? 3) METHODOLOGY - Maybe authors should better explain how they intend to reach aims previous exposed. - Just because author intend to compare urban area (Colombo) and rural area (Vavuniya), in the introduction they should mention the impact of urban/rural area in the onset of mental health. - page 10. In my opinion authors should put the synthesis of the source close to the letter (i.e. A existing twins, B maternity hospital, C field visit, D school) - It seems to me that in the table there are several characters mixed together. if so, please make the character uniform 4) DISCUSSION - In the conclusion section authors should mention also environmental factors (and not only genetic) for mental health problems. - Authors should provide some examples of confounding factors (lines 510-511) - In the discussion but in the whole text authors have never referred to neurodevelopmental disorders (DSM 5) and epigenetics.
--	---

REVIEWER	Baptiste Couvy-Duchesne The University of Queensland, Brisbane, Australia Brain and Spine Institute, Paris, France
REVIEW RETURNED	28-Apr-2019

GENERAL COMMENTS	Your initiative, to establish a child and adolescent twin register in Sri Lanka, is laudable, clever and highly needed for research and genetic research in particular. It feels that the authors have a clear vision of what precious data may be collected, from pre-birth all the way to late adolescence. I was especially impressed by the ambitious and clever design of recruitment. Finally, the authors' concern for ethical research collection should ensure long lasting engagement with the twins and the general population of Sri Lanka. I wish the authors all the best in establishing this great, and needed, resource and I look forward to seeing the first publications resulting from it. See minor comments below. 1) "A focus on Sri Lankan twins may limit the generalisability of some findings to other populations." Yes, but you can also frame this a positive – heritability is population specific and due to genetic differences between populations all ethnic groups, analyses need to be performed in all ethnic groups to understand a trait at a world-wide level. Being the first twin registry in Sri-Lanka will allow to answer questions that could not be studied before – and is playing towards genetic samples more representative of the human diversity. You could make the point that inhabitant of LMCI are often under-represented in genetic studies. See below a few references, but the issue is more and more discussed so you may find other/better references. https://www.cell.com/cell/fulltext/S0092-8674(19)30231-4 https://www.cell.com/trends/genetics/fulltext/S0168-9525(09)00185-1?_returnURL=https%3A%2F%2Flinkinghub.elsevier.com%2Fretrieve%2Fpii%2FS0168952509001851%3Fshowall%3Dtrue 2) I found the justification paragraph for twin samples a bit confusing, for example it was unclear to me what confounders you are referring to. I think that one of the main interest of twin models is to characterise a phenotypic association, and interrogate its nature (e.g. environmental or genetic). Thus, a phenotypic correlation may be separated into a genetic correlation and into an environmental component. In addition, the variance of a single trait may also be decomposed to measure how much of individual variation may be attributable to genetic differences by opposition to environmental sources of variance (familial or unique). See for example the reference book by Neale and Cardon (Methodology for Genetic Studies of Twins and Families). Or see these articles for example, that I find useful: https://www.ncbi.nlm.nih.gov/pubmed/22307698 https://www.nature.com/articles/hdy197810 You state all this clearly in the "Methodology" and "Discussion" sections, maybe I just missed the point you are trying to make in "Advantages of twin cohort studies" 3) Source population A: how many twins are you expected to reach by contacting the 4112 twin pairs of child bearing age? I guess that it depends on how more common are twins in DZ twins compared to the general population?
---

	Secondly, I am not sure I understood, are there 1000 twins from Colombo who are under 18? 4) L274-275, I think there is a typo (extra word)
--	---

VERSION 1 – AUTHOR RESPONSE

REVIEWER ONE COMMENTS AND RESPONSES

Comment: 1) The English language must be completely revised by a native speaker

Response: Done

Comment: in my opinion this section should be revised. More information should be provided: a) the multifactorial aetiology of mental health disorder that includes also environmental factors; b) the need of preventive actions on these environmental factors. Information are presented too briefly and disconnected from each other

Response: this section was revised as suggested

Multifactorial aetiology of mental disorders especially in the developmental age

Challenges to early brain development take place in intra uterine life and continue throughout childhood and adolescence. Nutrients and growth factors regulate brain development during foetal and early postnatal life and rapidly developing brain is more vulnerable to nutrient insufficiency. (Michael & Georgieff 2007) Low birth weight (LBW) and hypoxia are among the environmental factors most reliably associated with schizophrenia.

Poverty and adversities such as mother's ill health and high stress during pregnancy; early loss of parents via death or abandonment; witnessing inter-parental violence; dysfunctional parenting (particularly 'affectionless over control'); parental substance misuse, mental health problems and criminal behaviour; childhood sexual, physical and emotional abuse; childhood emotional or physical neglect; bullying; childhood medical illness; and war trauma have also been found to influence outcome. 2,3.

The consequences of disasters are well known causes of a variety of psychological and psychiatric disorders (Green 2003). Children and adolescents exposed to war related traumatic events are affected mental health distress, and in certain cases long term psychopathology (Betancourt & Khan 2008). There is also a strong relationship between adolescents exposed to trauma and suffering from PTSD and substance abuse. Studies have shown that more than half of the young people with PTSD subsequently develop substance abuse problems (Giaconia et al. 2000).

Prolonged adolescent major depressive disorder left untreated may reoccur in later adult life (Patton et al. 2014). However diagnosis and treatment of depression during adolescence is beneficial in reducing adult depression symptoms (Neufeld et al. 2017). Mental health disorders which develop during adolescence can have negative consequences which may result in increased risky decision making in later life (Hallfors et al. 2004; Shrier et al. 2001).

Social support is essential to young people for coping and protecting against the negative effects of stressors and for the promotion of positive mental health outcomes (Hussong 2000; Newman et al. 2007). Research findings indicate an inverse relationship between social support and depression. (Peirce et al. 2000; Newman et al. 2007)

Some of these adversities have been shown to be intergenerational, so that parents who themselves suffered in childhood struggle to provide an optimum environment for their own children. It is very likely that these types of events have their impact in interaction with other factors such as genetic predisposition and epigenetic processes.³ both gene-environment interaction and gene-environment covariation models have been proposed as explanations (Jennifer et al, 2013).

Hence, there is a crucial need to prioritize global health research directed at neurological, mental health and substance-use disorders in adolescents (Davidson et al, 2015), and there is a need as well as potential for preventive actions on these environmental factors.

Comment: page 6 line 85-86; Authors stated that reasons may include... but they cited only one reason "the stigma", in my opinion they have to discuss other factors

Response; amended section now stands as : Worldwide, 10-20% of children and adolescents experience mental disorders, and many transition into adulthood undiagnosed and continue to experience these disorders [1]. Reasons may include the lack of detection in schools and within families due to associated stigma,[2]. inadequate social support which has an inverse relationship with depression and is essential to young people for coping (Newman et al., 2007), children and adolescents being exposed to violence/trauma [4], looking from a socioecological model in which an adolescent with a particular genome undergoes physical and hormonal transformations within the context of a family, peers, school, work, community and culture all of which can be instrumental in determining adolescent NMS outcomes. Nature 527, S161-S166 (19 November 2015), DOI: 10.1038/nature16030), and even though faced high levels of mental health-related challenges, treatment and prevention resources for CAMH are generally low, even compared to adults (WPA, WHO, & IACAPAP,2005).

Comment: page 6 line 88, insert also impact of mental illness on well being

Response: done

Comment: -lines 83-93. I suggest to revise this section. I will first proceed by stating that the etiopathogenesis of mental disorders in the developmental age is multifactorial, then I will go on to say that in the LI countries it is worth investigating environmental factors (different from HI countries) and then I would mention the information of the WHO

Response: done

Comment: lines 94-100. Authors affirmed that studies in adolescence are scarce. In my opinion authors should reinforce this affirmation saying why adolescence is important (the onset of several mental disorders is in adolescence; adolescence is a critical developmental psychological period)

Response: added Adolescence is a critical developmental psychological period. Mental health is a key factor in the overall health and wellbeing of a child. Prolonged adolescent major depressive disorder left untreated may reoccur in later adult life [8]

Comment: lines 101-108. Authors should introduce the concept of resilience and then use the term

Response: Done. This paragraph now read as; In general, resilience has been defined and described as ability in children and adults to adapt to, adjust to or overcome chronic or acute adversity, providing protection against the development of psychopathology. (Bonanno G. Loss, trauma, and human resilience: have we underestimated the human capacity to thrive after extremely aversive events? Am Psychol 2004; 59:20–8). It is a multidimensional characteristic that may vary with settings, situations,

socio-demographical characteristics, and individually at a more molecular and cellular level and the construct of resilience has been explored in biological, psychosocial, genetic and neurobiological fields [18]. Sri Lanka which has been through many hardships is a resilient nation; proven through low prevalence of common mental disorders including PTSD during a 3 decade civil war [19]. However, research on resilience is still scarce in LMIC. Therefore it is important to identify enabling factors and follow the life-course of resilience of children and adolescents for intervention strategy and policy formulation.

Comment: -The impact of the challenges to early brain development on mental health and on the etiology of mental illness. I do not understand this paragraph. If author would expose neurodevelopment they should refer not only to pregnancy period and perinatal period and not only to schizophrenia. Rationale is that neurodevelopment is not only intrauterine or perinatal period and not only linked to schizophrenia. The chapter is really incomplete and not linked with background and subsequent paragraphs. I believe that if the authors want to keep this paragraph they must proceed with an extensive narrative revision of the topic otherwise it is better to delete the section

Response: deleted

Comment: Why cohort studies?. Could author briefly explain the characteristics of cohort studies?

Response: Cohort studies in which groups of people are followed up over time are the best method to determining disease incidence and the natural history of a condition

Comment:

3) METHODOLOGY

- Maybe authors should better explain how they intend to reach aims previous exposed.
- Just because author intends to compare urban area (Colombo) and rural area (Vavuniya), in the introduction they should mention the impact of urban/rural area in the onset of mental health.
- page 10. In my opinion authors should put the synthesis of the source close to the letter (i.e. A existing twins, B maternity hospital, C field visit, D school)
- It seems to me that in the table there are several characters mixed together. if so, please make the character uniform

Response: Not clear what the reviewer mean here ' Maybe authors should better explain how they intend to reach aims previous exposed'.

All others done

The majority of the Sri Lankan population live in the rural sector (77.4%); the urban population share is 18.2%, while 4.4% live in large agricultural estates, considered a separate sector. Urbanization is relatively high in the Western province (38.8 %) and very low in the North Central (4.0%) and North Western (4.1%) provinces. Distribution of facilities and human resources is unequal and specialized services are relatively less in rural areas. Access to health care facilities is more difficult due to greater distances and limited transport services. Poverty is a major problem and education levels are relatively lower in rural areas. Vavuniya has been chosen as a comparative setting as it's a more rural region which was affected by the conflict and war.

There can be an impact of urban/rural differences in the onset of mental health, and therefore we included these two areas to develop the register initially.

Comment

4) DISCUSSION

- In the conclusion section authors should mention also environmental factors (and not only genetic) for mental health problems.

Response:

- Authors should provide some examples of confounding factors (lines 510-511)

- In the discussion but in the whole text authors have never referred to neurodevelopmental disorders (DSM 5) and epigenetics.

Reviewer two

Comment

1) "A focus on Sri Lankan twins may limit the generalisability of some findings to other populations."

Yes, but you can also frame this a positive – heritability is population specific and due to genetic differences between populations all ethnic groups, analyses need to be performed in all ethnic groups to understand a trait at a world-wide level. Being the first twin registry in Sri-Lanka will allow to answer questions that could not be studied before – and is playing towards genetic samples more representative of the human diversity.

You could make the point that inhabitants of LMCI are often under-represented in genetic studies. See below a few references, but the issue is more and more discussed so you may find other/better references.

[https://www.cell.com/cell/fulltext/S0092-8674\(19\)30231-4](https://www.cell.com/cell/fulltext/S0092-8674(19)30231-4)

[https://www.cell.com/trends/genetics/fulltext/S0168-9525\(09\)00185-](https://www.cell.com/trends/genetics/fulltext/S0168-9525(09)00185-1?_returnURL=https%3A%2F%2Flinkinghub.elsevier.com%2Fretrieve%2Fpii%2FS0168952509001851%3Fshowall%3Dtrue)

[1?_returnURL=https%3A%2F%2Flinkinghub.elsevier.com%2Fretrieve%2Fpii%2FS0168952509001851%3Fshowall%3Dtrue](https://www.cell.com/trends/genetics/fulltext/S0168-9525(09)00185-1?_returnURL=https%3A%2F%2Flinkinghub.elsevier.com%2Fretrieve%2Fpii%2FS0168952509001851%3Fshowall%3Dtrue)

Response: all done and the paragraph now stand as

A focus on Sri Lankan twins may limit the generalisability of some findings to other populations. However, heritability is population specific and due to genetic differences between populations all ethnic groups, analyses need to be performed in all ethnic groups to understand a trait at a world-wide level. Being the first twin registry in Sri-Lanka will allow to answer questions that could not be studied before – and is playing towards genetic samples more representative of the human diversity. Inhabitants of LMCI are often under-represented in genetic studies and therefore the studies arising from this data base will add significant value to the research in LMIC (ref).

Comment

2) I found the justification paragraph for twin samples a bit confusing, for example it was unclear to me what confounders you are referring to. I think that one of the main interests of twin models is to characterise a phenotypic association, and interrogate its nature (e.g. environmental or genetic). Thus, a phenotypic correlation may be separated into a genetic correlation and into an environmental component. In addition, the variance of a single trait may also be decomposed to measure how much of individual variation may be attributable to genetic differences by opposition to environmental sources of variance (familial or unique). See for example the reference book by Neale and Cardon (Methodology for Genetic Studies of Twins and Families). Or see these articles for example, that I find useful: <https://www.ncbi.nlm.nih.gov/pubmed/22307698>

<https://www.nature.com/articles/hdy197810>

You state all this clearly in the "Methodology" and "Discussion" sections, maybe I just missed the point you are trying to make in "Advantages of twin cohort studies"

Response: This paragraph now stand as,

Advantages of twin cohort studies

In longitudinal observational studies, confounding poses a considerable threat to the validity of studies aiming to identify causal mechanisms. Twin cohorts can be useful as they allow researchers to understand whether phenotypic relationships are underpinned by shared genetic and or shared environments. Moreover, they can also look at the etiology of 'environmental variables' and the etiological relationship between these environments and outcomes of interest. A number of studies have found evidence of genetic influences on environmental variables such as stressful life events (Kender et al 2007) – a phenomena know as gene-environment correlation (Scarr & McCartney 1983). This additional etiological information can help us to understand more about the causal relationships phenotypes and environments and phenotypes of interest.

As twinning is largely unrelated to environmental risk factors such as poverty or social class, they allow generalizable assessments of associations and the ability to evaluate the extent of both genetic and environmental confounding; one of the reasons for the increased popularity of twin studies over the past decades,[23].

Comment 3) Source population A: how many twins are you expected to reach by contacting the 4112 twin pairs of child bearing age? I guess that it depends on how more common are twins in DZ twins compared to the general population?

Secondly, I am not sure I understood, are there 1000 twins from Colombo who are under 18?

Response

Source population A (Existing twins from adult twin registry)

Twin off-spring are more common among dizygotic twins compared with singletons and monozygotic pairs (Hoekstra et al 2008). The adult population based twin register in Colombo comprises 9500 twin pairs of which 4112 twin pairs are of child bearing age. Of the 4112 approximately 2% would be predicted to have DZ twins (N=82). There are also around 1000 pairs from Colombo up to the age 18 voluntarily registered since establishing the adult register. A dedicated team of research assistants will contact these adult twins by phone and postal methods to determine twin off-springs aged up to 18 years and invite them to register. Ethics approval and consent has been obtained previously for contacting these twins.

(ref <https://ghr.nlm.nih.gov/primer/traits/twins>)

Yes there are around 1000 pairs of twins under the age of 18 years in the existing register

4) L274-275, I think there is a typo (extra word)

Done

VERSION 2 – REVIEW

REVIEWER	Gennaro Catone Suor Orsola Benincasa University
REVIEW RETURNED	30-Jun-2019

GENERAL COMMENTS	I do not find the paper improved. The article is not easy to read, the introduction is made up of scattered contributions absolutely not linked to each other. The rest of the article and above all the objectives and the methodology set out are valid but in general the quality of the article is insufficient. My advice is to completely rewrite the article with a more streamlined introduction, continuing to highlight only the main points of the work. Being a project presentation article, I recommend a brief report.
---

REVIEWER	Baptiste Couvy-Duchesne The University of Queensland, Brisbane, Australia
REVIEW RETURNED	26-Jun-2019

GENERAL COMMENTS	Thank you for the revisions. I have no more comments.
---

VERSION 2 – AUTHOR RESPONSE

Reviewer: 1 (Gennaro Catone)

They have completely rewritten the introduction section. Therefore, instead of track changes we have bolded the text in the introduction as per guidelines of the journal. However, track changes in the other parts of the manuscript remain as per guidelines.

VERSION 3 – REVIEW

REVIEWER	Gennaro Catone Suor Orsola Benincasa University Italy
REVIEW RETURNED	24-Aug-2019

GENERAL COMMENTS	the manuscript has been improved. Please correct any typographical error. wow
--